# Chimney vs. Fenestrated Endovascular vs. Open Repair for Juxta/Pararenal Abdominal Aortic Aneurysms: Systematic Review and Network Meta-Analysis of the Medium-Term Results

**DOI:** 10.3390/jcm11226779

**Published:** 2022-11-16

**Authors:** Petar Zlatanovic, Aleksa Jovanovic, Paolo Tripodi, Lazar Davidovic

**Affiliations:** 1Clinic for Vascular and Endovascular Surgery, University Clinical Centre of Serbia, 11000 Belgrade, Serbia; 2Institute of Epidemiology, Faculty of Medicine, University of Belgrade, 11000 Belgrade, Serbia; 3Vascular Surgery Division, Hospital Clinic Universitari Sagrat Cor, University of Barcelona, 08007 Barcelona, Spain; 4Faculty of Medicine, University of Belgrade, 11000 Belgrade, Serbia

**Keywords:** abdominal aortic aneurysm (AAA), juxtarenal, pararenal, endovascular aneurysm repair (EVAR), fenestrated EVAR (FEVAR), chimney EVAR (ChEVAR), open surgery, medium-term

## Abstract

**Introduction:** This systematic review with network meta-analysis aimed at comparing the medium-term results of open surgery (OS), fenestrated endovascular repair (FEVAR), and chimney endovascular repair (ChEVAR) in patients with juxta/pararenal abdominal aortic aneurysms (JAAAs/PAAAs). **Materials and methods:** MEDLINE, SCOPUS, and Web of Science were searched from inception date to 1st July 2022. Any studies comparing the results of two or three treatment strategies (ChEVAR, FEVAR, or OS) on medium-term outcomes in patients with JAAAs/PAAAs were included. Primary outcomes were all-cause mortality, aortic-related reintervention, and aortic-related mortality, while secondary outcomes were visceral stent/bypass occlusion/occlusion, major adverse cardiovascular events (MACEs), new onset renal replacement therapy (RRT), total endoleaks, and type I/III endoleak. **Results:** FEVAR (OR = 1.53, 95%CrI 1.03–2.11) was associated with higher medium-term all-cause mortality than OS. Sensitivity analysis including only studies that analysed JAAA showed that FEVAR (OR = 1.65, 95%CrI 1.08–2.33) persisted to be associated with higher medium-term mortality than OS. Both FEVAR (OR = 8.32, 95%CrI 3.80–27.16) and ChEVAR (OR = 5.95, 95%CrI 2.23–20.18) were associated with a higher aortic-related reintervention rate than OS. No difference between different treatment options was found in terms of aortic-related mortality. FEVAR (OR = 13.13, 95%CrI 2.70–105.2) and ChEVAR (OR = 16.82, 95%CrI 2.79–176.7) were associated with a higher rate of medium-term visceral branch occlusion/stenosis compared to OS; however, there was no difference found between FEVAR and ChEVAR. **Conclusions:** An advantage of OS compared to FEVAR and ChEVAR after mid-term follow-up aortic-related intervention and vessel branch/bypass stenosis/occlusion was found. This suggests that younger, low-surgical-risk patients might benefit from open surgery of JAAA/PAAA as a first approach.

## 1. Introduction

The complexity of abdominal aortic aneurysm (AAA) repair depends mainly on anatomical detail relating to the segment of non-dilated aorta between renal arteries and the aneurysm, referred to as the aneurysm ‘neck’. Some 40–60% of aneurysms fall within the category of infrarenal AAA with adequate neck characteristics [1,2], and there is a wealth of comparative effectiveness evidence relating to such patients [3]. Aneurysms that have a neck that is too short or otherwise unsuitable for standard endovascular aneurysm repair (EVAR) within instructions for use (IFUs) are referred to as “complex aneurysms”.

Juxta/pararenal abdominal aortic aneurysms (JAAAs/PAAAs) are a frequent variation of complex abdominal aortic aneurysms (AAAs). JAAA is defined by European Society for Vascular Surgery guidelines and Society for Vascular Surgery reporting standards as “AAA which extends up to renal arteries but does not involve them”, while PAAA is defined as “AAA where at least one or both renal arteries derive from AAA itself, but does not involve the superior mesenteric artery” [4,5,6].

Recent systematic reviews and meta-analyses have demonstrated the short-term benefit of fenestrated and chimney endovascular aneurysm repair (FEVAR/ChEVAR) in comparison with open surgery (OS) [7]; however, medium/long-term results are scarce. Previous reviews performed on the subject were either scoping in nature and lacked an analytical approach, or incorporated studies contributing data from only a single approach [8,9]. Others included also patients with suprarenal and paravisceral AAAs and thoracoabdominal AAAs [10].

This systematic review and network meta-analysis aimed at comparing the medium-term results of OS, FEVAR, and ChEVAR for patients with juxta/pararenal abdominal aortic aneurysms.

## 2. Methods

We performed a systematic review following the Preferred Reporting Items for Systematic Reviews and Meta-Analyses (PRISMA) and assessing the methodological quality of systematic reviews (AMSTAR) guidelines [11,12]. MEDLINE, SCOPUS, and Web of Science were searched from their inception to 1 July 2022 for studies reporting comparative outcomes for patients with JAAAs/PAAAs undergoing two or more treatment modalities: OS, FEVAR, OR ChEVAR. No restrictions were placed in terms of publication type. Grey literature was not searched. The full search strategy is available in Appendix A. The study was registered on PROSPERO on 11 August 2021 (record number CRD42021267189).

### 2.1. Screening and Study Selection

We included studies comparing the results of two or three treatment strategies (OS, FEVAR, and ChEVAR) on medium-term clinical outcomes for patients with JAAAs/PAAAs. Medium-term was defined as a follow-up period of at least 6 months, ranging up to 60 months. Studies with standard infrarenal AAAs, AAAs with long but hostile neck characteristics (excessive thrombus, excessive angulation, conical shape, or calcifications) where standard infrarenal EVAR outside instructions for use has been performed, infected AAAs, ruptured AAA patients, thoracoabdominal aortic aneurysms (ThAAAs), suprarenal AAAs involving visceral segment at the level of the superior mesenteric artery and celiac trunk, connective-tissue-related aneurysms, studies containing less than 10 patients per treatment arm (to ensure enough experience in the treatment of this complex AAA pathology), performed before 2010 (to reduce the number of different stent graft generations in the comparison), and those having less than 6 months of follow-up were excluded from the analysis. Systematic reviews with or without meta-analysis, traditional reviews, comments, editorials and letters, and case reports were excluded, as well as any animal studies. A hand search of systematic reviews was also performed. Non-English articles were excluded unless they had an English abstract with extractable data. Two reviewers (P.Z. and P.T.) independently screened titles and abstracts as well as full texts of potentially eligible studies. Disagreements were resolved by discussion or consulting the third and fourth authors (A.J. and L.D.). The Rayyan systematic review web application (Available from www.rayyan.ai, accessed on 12 August 2022) was used for abstract screening.

### 2.2. Data Extraction and Definitions

Two authors independently extracted data, and any disagreements were resolved by the third and fourth authors. The following data were extracted from each study: study characteristics, study demographics, and periprocedural data (Appendix A).

### 2.3. Outcomes Measures

Outcome measures were decided a priori. The mean follow-up time point was 31.4 months and was considered to be a medium-term interval.

Primary medium-term outcomes:

All-cause mortality, aortic-related reintervention, and aortic-related mortality.

Secondary medium-term outcomes:

Visceral stent/bypass occlusion/occlusion; major adverse cardiovascular events (MACEs) that were defined as a composite endpoint of cardiac death, myocardial infarction, coronary artery revascularisation, stroke, and hospitalisation because of heart failure; new onset renal replacement therapy (RRT); total endoleaks; and type I/III endoleak (including persistent gutter type Ia endoleak in ChEVAR group).

### 2.4. Quality Assessment

No RCTs that fulfilled the inclusion criteria were found after the screening of the manuscripts. The Risk of Bias in Non-randomized Studies—of Interventions (ROBINS-I) tool was used to assess the quality of included observational studies [13]. The Grading of Recommendation Assessment, Development, and Evaluation (GRADE) system was used to analyse the overall quality of evidence and strength of recommendation for each of the outcomes [14]. The quality of evidence can be rated as “high”, “moderate”, “low”, or “very low”. Two reviewers (P.Z. and A.J.) independently performed the methodological quality assessment using the GRADEpro GDT software (available from gradepro.org) and risk of bias summaries generated using the robvis web tool [15].

### 2.5. Statistical Analysis

A network meta-analysis (NMA) within a Bayesian framework was performed using WinBUGS14 software, with codes adapted from Dias et al. [16]. The parameters were estimated using the Markov Chain Monte Carlo (MCMC) method. Results are based on 50,000 iterations using three chains, with an initial (burn-in) chain of 20,000. Model fit was assessed using posterior mean residual deviances and deviance information criteria (DIC). The transitivity assumption was assessed by observing the distribution of pre-operative characteristics in the studies, as well as the study designs. Odds ratios (ORs) with 95% credible intervals (95%CrI) were computed. Sensitivity analysis was performed for all primary outcomes—including studies reporting exclusively on JAAA.

## 3. Results

A total of 1723 publications were identified, and after abstract screening, 63 were deemed relevant and read in full text. The network meta-analysis included 16 studies [17,18,19,20,21,22,23,24,25,26,27,28,29,30,31,32]. The PRISMA flow diagram for study selection is presented in Figure 1.

### 3.1. Study Characteristics

The characteristics of the studies included in the network meta-analysis are presented in Table 1. A total of 4369 patients were included, with 2581 undergoing OS, 1498 FEVAR, and 290 undergoing ChEVAR. Most studies included patients with JAAAs [17,18,19,20,22,23,24,25,26,28,29,30,31,32]—only one study reported on patients with JAAAs/PAAAs [21], and one reported outcomes for PAAA [27].

Table 2 shows the procedural data of the included studies. Most patients had suprarenal proximal clamp position in the OS group, while the proximal clamping time ranged from 22 to 48 min. Zenith (Cook Medical, Bloomington, IN, USA) was the most frequent stent graft manufacturer in the studies where FEVAR was performed, while it was Endurant (Medtronic, Minneapolis, MN, USA) for ChEVAR. The most commonly used bridging stent graft in FEVAR and chimney graft in the ChEVAR group was Advanta V12 (Atrium Medical, Hudson, NH, USA). The most commonly present design for FEVAR/ChEVAR was with two fenestrations/chimney stent grafts, while the mean number of fenestrations or chimney stent grafts in endovascular interventions ranged from 1.4 to 2.8. The range of the duration of follow-up was 6–60 months.

### 3.2. Quality of Included Studies and Choice of Model

The overall quality of studies, as assessed by the ROBINS-I tool, was deemed “low”, with 10 studies (62%) having being deemed as having either serious or critical risk of bias in one or more domains [17,18,20,21,22,24,25,26,27,31], and 6 (38%) assessed as having moderate risk of bias [19,23,28,29,30,32] (Figure 2). The GRADE quality of evidence for all outcomes is presented in Appendix A and ranged from “very low” to “moderate”.

Values of the deviance information criteria (DIC) were similar in both models for all primary outcomes. However, lower values of residual deviance (Dres) were observed for the random-effects (RE) model compared with the fixed-effect model (FE) in all analyses. Seeing how both the values of Dres and the fact that studies were observational and heterogeneous were an indication for using the RE model, it was chosen in order for the estimate to be more conservative (Appendix A).

### 3.3. Primary Outcomes

Fourteen studies reported on all-cause medium-term mortality. This NMA included 4229 patients, and 417 deaths were reported (9.8%) (Figure 3a). The unweighted pooled medium-term mortality rate was 8.1% for OS, 12.3% for FEVAR, and 14.3% for ChEVAR. The NMA results indicated that FEVAR (OR = 1.53, 95%CrI 1.03–2.11) was associated with higher medium-term mortality compared with OS (Table 3). The sensitivity analysis in which only studies with JAAA patients were included showed that FEVAR (OR = 1.65, 95%CrI 1.08–2.33) persisted to be associated with higher medium-term mortality when compared with OS (Appendix A).

Eleven studies reported on aortic-related reintervention as an outcome. A total of 1497 patients were included in this NMA, with 146 patients who underwent aortic-related reintervention (11.2%) (Figure 3b). The unweighted pooled aortic-related reintervention rate was 3.6% for OS, 17.1% for FEVAR, and 16.1% for ChEVAR. The NMA results showed that both FEVAR (OR = 8.32, 95%CrI 3.80–27.16) and ChEVAR (OR = 5.95, 95%CrI 2.23–20.18) were associated with a higher aortic-related reintervention rate than OS (Table 3), and this association persisted after sensitivity analysis including only JAAA (OR = 9.61, 95%CrI 3.44–44.22 for FEVAR; OR = 7.11, 95%CrI 2.06–32.67 for ChEVAR) (Appendix A). There was no difference between FEVAR and ChEVAR in terms of aortic-related reintervention rates, not even after sensitivity analysis for JAAA (Table 3, Appendix A).

Ten studies reported on aortic-related mortality. A total of 1150 patients were included in NMA, with 12 patients contributing to aortic-related mortality (1.1%) (Figure 3c). The unweighted pooled aortic-related mortality rate was 0.9% for OS, 0.8% for FEVAR, and 1.7% for ChEVAR. Results from NMA as well as from sensitivity analysis including only JAAA showed no difference between different treatment options in terms of aortic-related mortality (Table 3, Appendix A).

### 3.4. Secondary Outcomes

Results from NMA showed that FEVAR (OR = 13.13, 95%CrI 2.70–105.2) and ChEVAR (OR = 16.82, 95%CrI 2.79–176.7) were associated with a higher rate of medium-term visceral branch occlusion/stenosis compared to OS; however, there was no difference between FEVAR and ChEVAR in terms of this complication. No difference was found between the three treatment options in terms of renal replacement therapy and MACEs. When comparing FEVAR and ChEVAR, no difference was found regarding the total number of endoleaks and more malignant ones such as type I/III (Table 4).

## 4. Discussion

This NMA found only observational studies comparing medium-term outcomes of interventions for JAAA/PAAA, mostly of low quality. The results of this NMA showed that FEVAR had a higher mid-term mortality compared to OS. Both endovascular procedures had higher rates of aortic-related reintervention and side branch occlusion/stenosis compared to the OS group. When making a comparison between the endovascular techniques, no significant preferences for either FEVAR or ChEVAR were found for any of the outcomes.

In this meta-analysis, several attempts were made to ameliorate the limitations of the existing literature. Firstly, it is difficult to interpret the results of the existing meta-analyses due to anatomical heterogeneity of the patients [10,33]. We focused our attention on studies reporting outcomes for only JAAA/PAAA, thus excluding more complex AAAs such as suprarenal and ThAAAs. Secondly, meta-analyses usually focus their attention on two most commonly used treatment options, i.e., FEVAR vs. OS, neglecting ChEVAR as the treatment option that is often used in some centres as the first-line endovascular option for the treatment of JAAA/PAAA [33]. This has only limited value. Thirdly, most of the published meta-analyses included studies published earlier than 2010 [7,34]. This NMA included more updated publications, with 11/16 published after 2015. Fourthly, NMA has an advantage that it allows both direct and indirect comparison; thus, more data are incorporated in the final analysis, and a bigger scope of the picture is tackled, whereas a single pairwise sometimes offers a very fragmented picture due to its failure to incorporate indirect data in the comparison.

Current guidelines recommend that the choice of different techniques and options for the management of JAAA/PAAA in the elective setting should be considered based on patient status, anatomy, local routines, team expertise, and patient preference [4,5]. The findings of this NMA reconfirm the widely accepted observation that endovascular techniques are associated with a higher incidence of aortic-related reintervention and a higher incidence of branch stenosis/occlusion. Recommendation 96 in the ESVS guidelines states that “In complex endovascular repair of juxtarenal abdominal aortic aneurysm, endovascular repair with fenestrated stent grafts should be considered the preferred treatment option when feasible”. Current guidelines favour the advantage of endovascular techniques (FEVAR and ChEVAR) over OS in terms of short-term outcomes [4,5].

According to recommendation 97 from the latest ESVS guidelines [4], FEVAR is preferred over ChEVAR in the elective setting, while recommendation 98 says that ChEVAR might be used in the emergent setting as a bailout procedure. However, this recommendation is based on expert opinion, and there are no high-quality data that might support these two recommendations. One of the major concerns with ChEVAR in the elective setting is that it is associated with a high rate of type Ia endoleak, especially with more than two chimneys [35]. Furthermore, gutters created between the main graft and chimneys may limit the durability of the technique.

This NMA showed one interesting finding that FEVAR patients had worse medium-term all-cause mortality compared to the OS group. A similar tendency was found for ChEVAR patients compared to OS, but this did not reach the level of statistical significance. Due to the non-randomised nature of all studies included in this NMA, it is possible that this reflects a confounding from the variations in baseline clinical characteristics between the groups, but it is an important finding and an indication for future RCTs regardless. It is also possible that a confounding due to indication is present, i.e., that surgeons tend to choose endovascular solutions for less fit patients, and therefore, these solutions have worse medium-term outcomes. A general observation from Table 1 is that patients from the endovascular group were older, with the presence of other cardiovascular risk factors. Although the inclusion criteria are somewhat different, focusing also on patients with adverse neck characteristics, a recently published NMA from Patel et al. [36] showed no differences in overall mid-term mortality between three groups of patients.

A significantly higher rate of mid-term reinterventions in both the FEVAR and ChEVAR group compared to OS was demonstrated in this NMA. A recent NMA [36] showed that only FEVAR patients had a higher mid-term reintervention rate. It must be noted that details about OS reintervention are often lacking. One good example is the rate of postincisial hernia repair, which was reported only in one study, and it is unclear whether it was counted as reintervention. Another contributing factor could be reintervention due to persistent type II endoleak (seven studies) in the FEVAR group without mentioning specific reasons. Additionally, FEVAR nowadays for JAAA repair usually has four vessel fenestrations, and this more proximal/extensive repair predisposes patients for mid-term complications [6]. Nevertheless, more frequent reinterventions coupled with the higher costs of endovascular devices could raise an additional concern. The results from Michel M. et al. [37] showed that FEVAR is more expensive and not a cost-effective option for JAAA/PAAA at 2 years. Since this study was performed, new devices from different companies have been developed, which will hopefully decrease the cost of these stent grafts in the future. It is, however, important to emphasise that the majority of reports failed to report data adequately, thus introducing difficulties in data interpretation. Unlike rigid reporting systems in RCTs and prospective observational trials, retrospective studies do not provide an insight into the variability of surgeon preferences and department policies. One of the explanations why the reintervention rate was higher in the endovascular groups could be the higher incidence of branch vessel stenosis/occlusion and the presence of the non-negligible overall unweighted pooled 6.6% rate of type I/III (malignant) endoleak for FEVAR/ChEVAR groups.

The branch vessel occlusion/stenosis rate was lower in OSR. However, the interpretation of results should be taken with caution since most of the patients had juxtarenal AAA, and only 11.5% of all patients undergoing OSR had renal artery bypass/reattachment, which makes comparison to FEVAR/ChEVAR difficult. Surprisingly, there was no difference between FEVAR and ChEVAR in terms of incidence of new onset endoleaks and branch vessel patency. Most of the trials used company-manufactured stent grafts from Zenith Cook (Cook Medical, Bloomington, IN, USA) for FEVAR and from Endurant (Medtronic, Minneapolis, MN, USA) for ChEVAR. In the meta-analysis of Katsargyris et al. [8], no difference was observed for target vessel patency and short-term mortality for the treatment of JAAA. Additionally, no difference was observed regarding new onset RRT. However, the absence of difference in terms of these outcomes could be due to the lack of papers with sufficient power to highlight any statistical difference.

One important factor that should not be neglected is the impact of centre volume on patient outcomes. As reported in a previous registry, centres with high (>14) volume of open JAAA repair demonstrated significant adjusted lower perioperative mortality (3.9%) compared to centres with low volume of open repair [38]. Possibly, a broad implementation of centralisation of treatment of JAAA/PAAA could further improve these results, especially in OS.

## 5. Limitations and Implications for Research

There are several concerns in this NMA. Our study included only observational studies and registries with significant differences in terms of baseline clinical characteristics. No RCTs have been performed comparing JAAA/PAAA repair. The GRADE rating for evidence was “low” for the majority of pairwise comparisons, reflecting the inherited bias. Data veracity is the Achilles’ heel of all retrospective analyses. Such nuances were most apparent in big registries [22]. Another issue when analysing cohort studies as opposed to randomised controlled trials is confounding by indication. It is possible that frailer patients received endovascular treatment, while patients with better pre-operative conditions received OS. This confounding was impossible to account for in our analysis. Another cause of concern is the “learning curve bias” and the use of older generations of devices. We tried to avoid this issue by excluding studies that reported less than 10 patients per treatment arm and by only including studies that treated patients after 2010, but since the endovascular techniques have been further improved since then, it is possible that this bias is still present in our analysis. Furthermore, the role of physician-modified grafts and outside-of-use EVAR in the elective setting have not been investigated due to unstandardised use of these two techniques in the setting of JAAA/PAAA repair. There was a lack of standardisation of definitions and reporting of the anatomy. For example, some studies defined JAAA as a neck less than 10 mm, and others used the anticipated clamp site with no specific mention of whether the aneurysms had involved the renal artery ostia, i.e., PAAA. Although the analysis was focused on JAAA/PAAA, the majority of studies provide no detail on the AAA anatomy, such as neck length and other adverse features. Cost-effectiveness analysis and quality of life assessment were not performed and could be important outcomes that could help in the decision-making process between these patient groups.

## 6. Conclusions

The results of this NMA found an advantage for OS regarding aortic-related intervention and vessel branch/bypass stenosis/occlusion compared to FEVAR and ChEVAR after medium-term follow-up. This suggests that younger, low-surgical-risk patients might benefit from open surgery of JAAA/PAAA; however, this insight should be interpreted with caution due to the low quality of the included studies in the analysis and the possibility of confounding by indication, bearing in mind the observational design of the included studies. Further larger studies including experienced and high-volume AAA centres in patients with similar baseline patient characteristics are needed to adequately determine medium and long-term results of all three used treatment options.

## Figures and Tables

**Figure 1 jcm-11-06779-f001:**
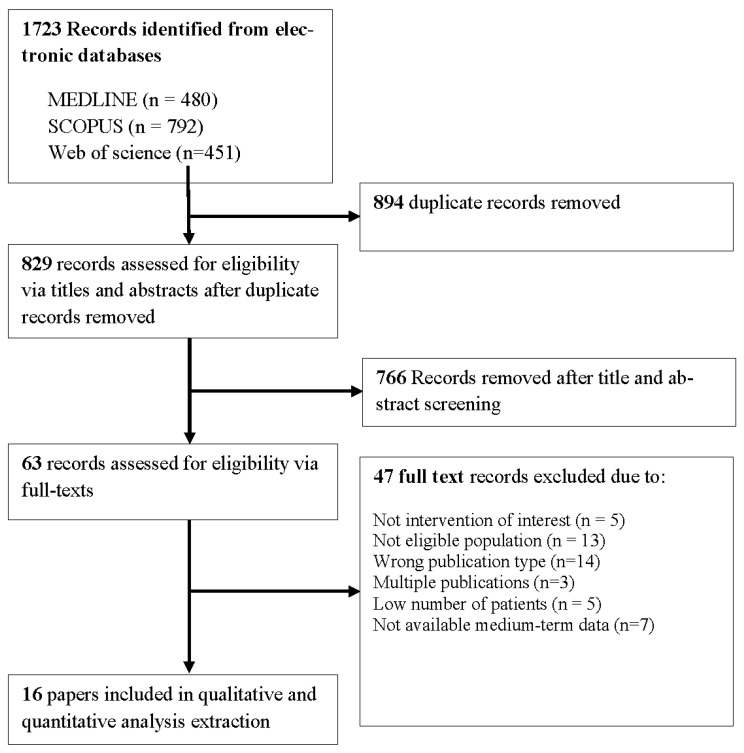
PRISMA flow diagram showing selection of included studies.

**Figure 2 jcm-11-06779-f002:**
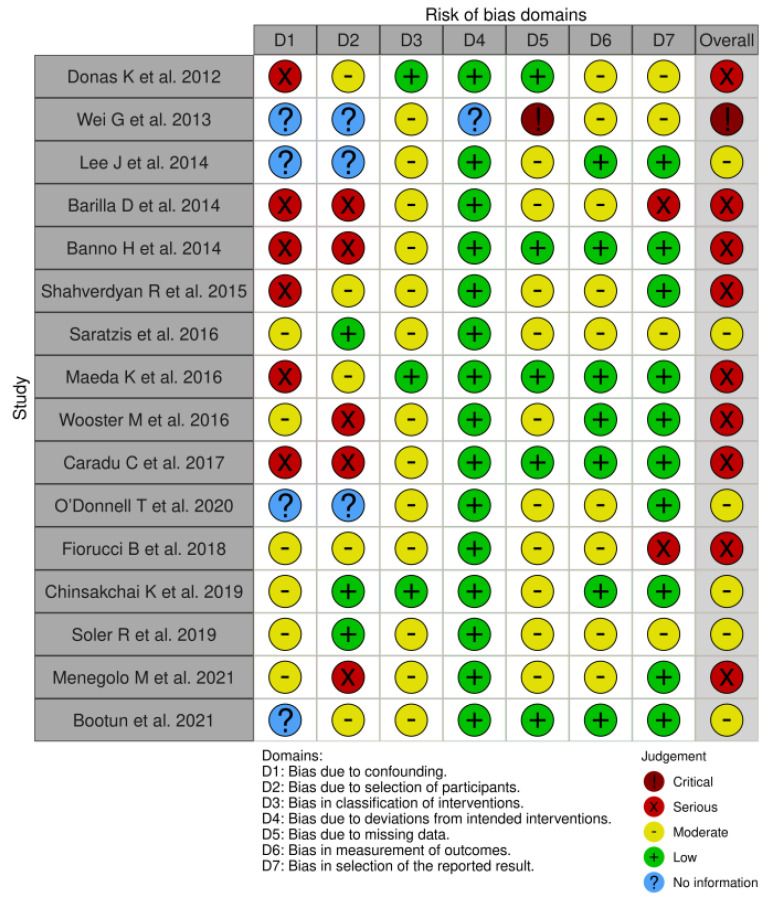
Risk of bias summary—judgments about each risk of bias item for each included study (for non-randomised studies using ROBINS-I tool) [17,18,19,20,21,22,23,24,25,26,27,28,29,30,31,32].

**Figure 3 jcm-11-06779-f003:**
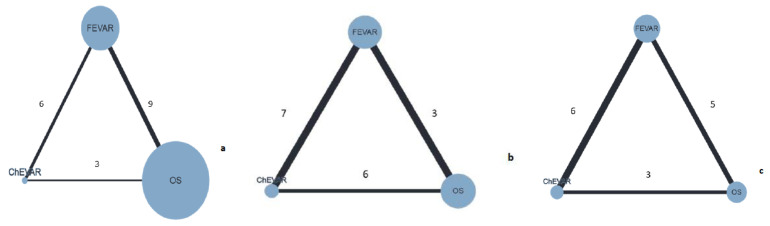
Network graph for mortality (**a**); aortic-related reintervention (**b**); and aortic-related mortality (**c**) (size of the node represents the sample size for the procedure, while the edge width represents the number of studies included in direct comparison).

**Table 1 jcm-11-06779-t001:** Baseline clinical characteristics of patients in included studies.

First Author (Year of Publication)	Study Design	AAA Type	Follow-Up Duration (Months)	Intervention	Sample Size	Mean AGE (years)	Female Gender (%)	Smoking (%)	HTN (%)	HLP (%)	Diabetes (%)	COPD (%)	CAD (%)	CVD (%)	CKD (%)	AAA Size (mm)	Previous Aortic Intervention
Donas et al. (2012) [17]	Retrospective observational	Juxtarenal	14.2	ChEVAR	30	74	10	-	-	-	-	33.3	33.3	-	23.3	62	11
				FEVAR	29	73	0	-	-	-	-	27.9	41.4	-	17.2	65	8
				Open surgery	31	71	12.9	-	-	-	-	19.3	12.9	-	6.4	60	2
Wei et al. (2013) [18]	Retrospective observational	Juxtarenal	12	ChEVAR	37	-	-	-	-	-	-	-	-	-	-	-	-
				FEVAR	13	-	-	-	-	-	-	-	-	-	-	-	-
Lee et al. (2014) [19]	Retrospective observational	Juxtarenal	6	ChEVAR	15	76	26.6	93.3	100	-	6.6	33.3	-	-	-	66	-
				FEVAR	15	77	33.3	73.3	93.3	-	33.3	13.3	-	-	-	61	-
Barilla et al. (2014) [20]	Retrospective observational	Juxtarenal	60	Open surgery	50	78	6	16	75	56	22	44	54	8	20	-	-
				FEVAR	50	71	4	22	75	50	24	48	58	6	16	-	-
Banno et al. (2014) [21]	Retrospective observational	Juxta/pararenal	24	FEVAR	80	74	10	-	70	51	17.5	32.5	55	16.2	15	58	4
				ChEVAR	38	74	10.5	-	79	58	26.3	29	39.5	10.5	23.7	66	9
Shahverdyan et al. (2015) [22]	Retrospective observational	Juxtarenal	24	Open surgery	34	72	23.5	-	76.5	-	14.7	38.2	53	-	5.9	-	-
				FEVAR	35	72	14.3	-	85.7	-	5.7	20	45.7	-	11.4	-	-
Saratzis et al. (2015) [23]	Retrospective observational	Juxtarenal	20	FEVAR	58	75	12	81	63.8	67.2	31	-	31	8.6	-	52	
				Open surgery	58	74	12	81	63.8	67.2	31	-	25.4	8.6	-	53	-
Maeda et al. (2015) [24]	Retrospective observational	Juxtarenal	33	Open surgery	81	71	14.8	72.8	74.1	40.7	24.7	23.4	54.3	24.7	22.2	58	-
				FEVAR	34	77	11.7	58.8	82.3	50	17.6	20.6	26.5	20.6	23.5	55	-
				ChEVAR	37	77	10.8	59.4	81.1	48.6	18.9	21.6	27	18.9	21.6	56	-
Wooster et al. (2016) [25]	Retrospective observational	Juxtarenal	15	ChEVAR	54	78	11.5	87	87	70.3	20.4	31.5	63	9.2	-	62	-
				FEVAR	39	72	12.8	79.5	82	66.6	15.4	38.5	53.8	17.9	-	68	-
Caradu et al. (2017) [26]	Retrospective observational	Juxtarenal	24	FEVAR	90	71	2.2	70	80	61.1	12.2	26.6	-	10	14.4	58	-
				ChEVAR	31	75	16.1	67.7	77.4	67.7	9.7	35.5	-	6.4	38.7	67	-
Fiorucci et al. (2018) [27]	Retrospective observational	Pararenal	50	FEVAR	92	75	9.8	-	84.8	32.6	11.9	41.3	57.6	10.8	18.5	-	-
				Open surgery	108	70	5.5	-	88.9	39.8	14.8	51.8	30.5	1	30.5	-	-
Chinsakchai et al. (2019) [28]	Retrospective observational	Juxtarenal	37	Open surgery	32	70	21.9	15.6	81.2	46.9	15.6	3.1	46.9			63	-
				FEVAR	20	72	15	15	85	15	20	15	30			55	-
				ChEVAR	23	76	34.8	4.3	65.2	30.4	8.7	8.7	30.4			68	-
Soler et al. (2019) [29]	Retrospective observational	Juxtarenal	27	Open surgery	134	69	7.4	86.5	67.9	61.2	14.2	33.6	40.3	-	20.9	59	5
				FEVAR	57	74	5.2	84.2	78.9	59.6	12.3	56.1	49.1	-	22.8	55	9
O’Donnell et al. (2020) [30]	Retrospective observational	Juxtarenal	36	Open surgery	1894	70	21	47	85	-	16	33	37	-	34	-	0
				FEVAR	822	73	20	34	86	-	19	38	45	-	42	-	6
Menegolo et al. (2021) [31]	Retrospective observational	Juxtarenal	43	ChEVAR	25	77	28	-	100	-	25	32	40	-	40	67	8
				Open surgery	61	71	3.3	-	82	-	14.7	9.8	29.5	-	22.9	65	0
Bootun et al. (2021) [32]	Retrospective observational	Juxtarenal	60	Open surgery	98	74	19	-	19.4	-	11.2	22.4	32.6	4.1	16.3	67	-
				FEVAR	64	76	4	-	6.2	-	9.4	25	45.3	20.3	18.7	76	-

AAA: abdominal aortic aneurysm;; FEVAR: fenestrated EVAR; ChEVAR: chimney EVAR; HTN: Hypertension; HLP: hyperlipidaemia; COPD: chronic obstructive pulmonary disease; CAD: coronary artery disease; CVD: cerebrovascular disease; CKD: chronic kidney disease.

**Table 2 jcm-11-06779-t002:** Procedural data of included studies.

First Author (Year of Publication)	AAA Type	Intervention	Sample Size	Stent Graft Manufacturer	Chimney/Bridging Stent Graft Manufacturer	FEVAR/ChEVAR Configuration					Open Surgery					Operation Duration (min)	Blood Loss (ml)	ICU Stay (Days)	Hospital Stay (Days)
						One Fenestration/One Chimney	Two Fenestrations/Two Chimneys	Three Fenestrations/One Chimneys	Four Fenestrations/One Chimneys	Mean Number of Fenestrations/Chimneys	Infrarenal Clamp	Interrenal Clamp	Suprarenal Clamp	Supraceliac Clamp	Mean Proximal Clamp Duration (min)				
Donas et al. (2012) [17]	Juxtarenal AAA	Chimney EVAR	30	Medtronic	Advanta balloon expandable	22	5	3	0	1.4	-	-	-	-	-	90	-	-	3
		FEVAR	29	Cook Zenith	-	0	29	0	0	2	-	-	-	-	-	290	-	-	3
		Open surgery	31	-	-	-	-	-	-	-	-	-	-	-	23	-	-	-	7
Wei et al. (2013) [18]	Juxtarenal AAA	Chimney EVAR	37	-	-	-	-	-	-	-	-	-	-	-	-	-	-	-	-
		FEVAR	13	-	-	-	-	-	-	-	-	-	-	-	-	-	-	-	-
Lee et al. (2014) [19]	Juxtarenal AAA	Chimney EVAR	15	Medtronic/Gore/Cook Zenith	iCAST and Viabahn	-	-	-	-	-	-	-	-	-	-	218	400	1	4
		FEVAR	15	Cook Zenith	iCAST	-	-	-	-	-	-	-	-	-	-	282	650	1	4
Barilla et al. (2014) [20]	Juxtarenal AAA	Open surgery	50	-	-	-	-	-	-	-	35	-	15	0	48	-	-	5	12
		FEVAR	50	-	-	2	48	0	0	1.9	-	-	-	-	-	-	-	3	12
Banno et al. (2014) [21]	Juxta/pararenal AAA	FEVAR	60	Cook Zenith, Anaconda	BECSs: Advanta V12 Fluency	3	44	29	4	2.4	-	-	-	-	-	191	-	-	8
		Chimney EVAR	38	-	BECSs, Advanta V12 Fulency	20	14	4	0	1.6	-	-	-	-	-	183	-	-	2
Shahverdyan et al. (2015) [22]	Juxtarenal AAA	Open surgery	34	-	-	-	-	-	-	-	0	15	14	5	-	171	-	-	11
		FEVAR	35	Cook Zenith, Anaconda	Advanta V12	2	19	11	3	2.3	-	-	-	-	-	188	-	-	7
Saratzis et al. (2016) [23]	Juxtarenal AAA	FEVAR	58	Cook Zenith	-	3	23	27	5	2.6	-	-	-	-	-	-	-	-	6
		Open surgery	58	-	-	-	-	-	-	-	-	-	-	-	-	-	-	-	7
Maeda et al. (2016) [24]	Juxtarenal AAA	Open surgery	81	-	-	-	-	-	-	-	0	28	47	6	35	313	3120	-	15
		FEVAR	34	Cook Zenith	Advanta V12	-	-	-	-	-	-	-	-	-	-	282	550	-	8
		Chimney EVAR	37	Gore Excluder	Advanta V12	-	-	-	-	-	-	-	-	-	-	198	440	-	8
Wooster et al. (2016) [25]	Juxtarenal AAA	Chimney EVAR	54	Medtronic, Gore Excluder, Cook Zenith	-	27	21	4	2	1.6	-	-	-	-	-	233	634	2	6
		FEVAR	39	Cook Zenith	-	1	16	21	1	2.5	-	-	-	-	-	238	408	2	6
Caradu et al. (2017) [26]	Juxtarenal AAA	FEVAR	90	Cook Zenith, Anaconda	Advanta 12	0	72	14	4	2.2	-	-	-	-	-	182	-	1	7
		Chimney EVAR	31	Cook Zenith, Medtronic	Fluency	-	-	-	-	1.3	-	-	-	-	-	139	-	1	7
Fiorucci et al. (2018) [27]	Pararenal AAA	FEVAR	92	Cook Zenith	-	8	30	29	25	2.8	-	-	-	-	-	218	344	-	10
		Open surgery	108	-	-	-	-	-	-	-	-	-	-	-	-	237	758	-	9
Chinsakchai et al. (2019) [28]	Juxtarenal AAA	Open surgery	32	-		-	-	-	-	-	3	9	10	10	-	242	-	1	8
		FEVAR	20	Cook Zenith	Advanta V12	-	-	-	-	2.4	-	-	-	-	-	262	-	1	7
		Chimney EVAR	23	-	Advanta V12, BeGraft, Viabahn	-	-	-	-	1.8	-	-	-	-	-	270	-	1	9
Soler et al. (2019) [29]	Juxtarenal AAA	Open surgery	134	-	-	-	-	-	-	-	0	10	91	33	23	-	-	1	11
		FEVAR	57	Cook Zenith	Blue Genesis, Advanta V12	3	25	22	7	2.6	-	-	-	-	-	-	-	2	8
O’Donnell et al. (2020) [30]	Juxtarenal AAA	Open surgery	1894	-	-	-	-	-	-	-	-	-	-	-	-	-	-	-	-
		FEVAR	822	-	-	-	-	-	-	-	-	-	-	-	-	-	-	-	-
Menegolo M et al. (2021) [31]	Juxtarenal AAA	Chimney EVAR	25	Medtronic, Gore Excluder, Cook Zenith	Advanta V12, Viabahn, Fluency	16	8	0	1	1.4	-	-	-	-	-	244	-	1	10
		Open surgery	61	-	-	-	-	-	-	-	0	28	21	12	22	214	-	3	8
Bootun et al. (2021) [32]	Juxtarenal AAA	Open surgery	98	-	-	-	-	-	-	-	0	43	55	0	24	-	1600	3	10
		FEVAR	64	-	-	4	22	23	15	2.7	-	-	-	-	-	-	-	1	6

AAA: abdominal aortic aneurysm; EVAR: endovascular aneurysm repair; FEVAR: fenestrated EVAR; ChEVAR: chimney EVAR.

**Table 3 jcm-11-06779-t003:** Network meta-analysis of major long-term outcomes in patients undergoing repair of JAAA/PAAA.

	FEVAR vs. OS	ChEVAR vs. OS	ChEVAR vs. FEVAR	Heterogeneity
Mortality	1.53 (1.03–2.11)	1.35 (0.74–2.40)	0.89 (0.50–1.58)	0.23 (0.01–0.71)
Aortic-related reintervention	8.32 (3.80–27.16)	5.95 (2.23–20.18)	0.72 (0.28–1.55)	0.63 (0.04–1.63)
Aortic-related mortality *	0.65 (0.06–5.67)	0.99 (0.07–9.76)	1.55 (0.20–11.06)	0.89 (0.04–1.93)

Legend: values are presented as OR (95% CrI), the treatment stated first is the reference treatment, OR < 1 favours the first treatment; OS—open surgery; FEVAR—fenestrated endovascular repair; ChEVAR—chimney endovascular repair; JAAA—juxtarenal abdominal aortic aneurysm; PAAA—pararenal abdominal aortic aneurysm. * The studies by Donas K et al. [17], and Wei G et al. [18], and Soler R et al. [29] were excluded from the quantitative analysis, as they had 0 outcomes in both groups.

**Table 4 jcm-11-06779-t004:** Network meta-analysis of secondary long-term outcomes in patients undergoing repair of JAAA/PAAA.

	FEVAR vs. OS	ChEVAR vs. OS	ChEVAR vs. FEVAR	Heterogeneity
Long-term branch/bypass occlusion/stenosis	13.13 (2.701–105.2)	16.82 (2.79–176.7)	1.28 (0.34–5.11)	1.44 (0.39–1.97)
Renal replacement therapy	1.27 (0.13–13.87)	1.09 (0.02–48.97)	0.82 (0.02–32.71)	1.43 (0.13–1.98)
MACEs	1.57 (0.52–5.88)	6.96 (0.70–103.0)	4.39 (0.49–51.21)	0.49 (0.02–1.81)
Total endoleaks	/	/	1.14 (0.44–3.51)	0.84 (0.06–1.88)
Type I/III endoleaks	/	/	1.59 (0.52–5.43)	0.77 (0.05–1.88)

Legend: values are presented as OR (95% CrI), the treatment stated first is the reference treatment, OR < 1 favours the first treatment; OS—open surgery; FEVAR—fenestrated endovascular repair; ChEVAR—chimney endovascular repair; JAAA—juxtarenal abdominal aortic aneurysm; PAAA—pararenal abdominal aortic aneurysm.

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
