# Peer review of "Chimney vs. Fenestrated Endovascular vs. Open Repair for Juxta/Pararenal Abdominal Aortic Aneurysms: Systematic Review and Network Meta-Analysis of the Medium-Term Results"

_jcm, 2022, doi:10.3390/jcm11226779_

Round 1
Reviewer 1 Report
The authors performed a systematic review and network meta-analysis with the aim to compare mid-term outcomes after OR vs FEVAR vs ChEVAR.
There are however some major concerns with the study design that should be considered.
1. The authors aim to compare mid-term results, yet not only studies with mid-term results are included, hence comparison is potentially biased. The authors should consider to define mid-term, and use this as an inclusion criterion.
2. Transitivity and heterogeneity are major concerns in this study design. Especially considering the low-quality of the included studies. In the methods, the authors describe to assess transitivity, yet in the results, it is not mentioned. Were study patients comparable between the groups? Is the network-approach valid?
3. It should be considered that frailer patients receive endovascular treatment while fitter patients receive OR (this concern has been raised by the authors, but is a major bias for the outcomes). This may explain the difference in mortality and should be mentioned in the conclusion (next to other biases).
Author Response
We would like to thank the reviewer for the thorough and dedicated work they have put into reviewing our manuscript. We believe that their observations have improved the quality of our manuscript, and for this we would also like to thank them.
Reviewer 1:
The authors performed a systematic review and network meta-analysis with the aim to compare mid-term outcomes after OR vs FEVAR vs ChEVAR.
ANS: We thank the reviewer for taking the time to review our manuscript.
There are however some major concerns with the study design that should be considered.
- The authors aim to compare mid-term results, yet not only studies with mid-term results are included, hence comparison is potentially biased. The authors should consider to define mid-term, and use this as an inclusion criterion.
ANS: Thank you very much for your comment. We considered papers that had at least 6 months follow-up period. Our initial idea was to analyze long-term data, but there were no papers that had a follow-up period longer than 60 months. That is why we were left with mid-term data, ranging from 6 to 60 months. We will insert this definition in the methodology section, but cannot say that this is one of the inclusion criteria since there were no studies with long-term data.
- Transitivity and heterogeneity are major concerns in this study design. Especially considering the low-quality of the included studies. In the methods, the authors describe to assess transitivity, yet in the results, it is not mentioned. Were study patients comparable between the groups? Is the network-approach valid?
ANS: We thank the reviewer for this very important comment. Previously published meta-analyses have either included anatomically heterogeneous patients in the analysis (suprarenal, paravisceral AAA, and thoracoabdominal AAAs along with pararenal and juxtarenal), or failed to incorporate all three modalities because of using the pairwise approach. We feel that we have improved on the transitivity assumption in regards to prior studies by restricting our analysis to only patients with either pararenal or juxtarenal AAA, and improved further by doing a sensitivity analysis including only juxtarenal AAAs. The age, as the major confounder is roughly similar across the studies. Data is on other variables missing from many studies, so it is impossible to claim with absolute certainty, but we feel, bearing in mind all previously mentioned, that it was justified to perform NMA on this occasion.
- It should be considered that frailer patients receive endovascular treatment while fitter patients receive OR (this concern has been raised by the authors, but is a major bias for the outcomes). This may explain the difference in mortality and should be mentioned in the conclusion (next to other biases).
ANS: We thank the reviewer for bringing up this important bias to our attention. We have added the following to the limitations: “Another issue when analyzing cohort studies as opposed to randomized controlled trials is confounding by indication. It is possible that frailer patients received endovascular treatment, while patients with better pre-operative condition received OR. This confounding was impossible to account for in our analysis.” And the following was added to conclusions: “...and the possibility of existence of confounding by indication, bearing in mind the observational design of the included studies.”
Reviewer 2 Report
Congratulations to the authors on this review comparing the medium-term outcomes of 3 treatment options for juxta- and pararenal aneurysms. The work is interesting and methodically sound; however, I have several questions and remarks.
Paragraph 1/Introduction:
The second sentence seems a bit unclear. What do you mean by „this category“ and „such patients“? Juxtarenal AAA? Hostile necks? Please clarify and rephrase.
Second paragraph: „JAAA defined by the ESVS….“ „are“ is missing.
Methods
Paragraph 2.1: What do you exactly mean by „Studies with standard, ruptured AAA…. were excluded from the analysis“? Do you mean infrarenal AAA? Were all infrarenal AAA excluded or only ruptured infrarenal AAA? What about emergent cases? And what AAA with a long enough but otherwise hostile neck, i.e. thrombus, calfication, conical shape, excessive angulation – I guess these were excluded as out of IFU? I guess that for OSR infected aneurysms were also excluded? Please specify your inclusion/exclusion criteria.
Even by excluding studies published before 2010 you will still have different stentgraft generations!
How may visceral/renal bypasses in the OSR patients were there actually? Do the included studies report on this in detail? In my experience it is much more common to incorporate the renal arteries directly into the proximal anastomosis in open JAAA repair than by selective bypass. Therefore, the comparison to target vessel complications in FEVAR/CHEVAR is difficult. This aspect should also be added to your discussion.
Paragraph 2.3:
How was MACE defined?
The main issue with CHEVAR are gutter endoleaks. How were they defined? Did you count them as Type Ia? Please specify since this has impact on your results.
Figure 1: You mention full text studies excluded due to lack of long term data – should this not be lack of midterm data since you report on midterm results?
Results
Paragraph 3.1:
Please add complete manufacturer data of all manufacturers (company, city, state/country).
Is there information in the included studies on wether any kind of renal protection (i.e. cold perfusion) was used in the patients with suprarenal clamping? Were there also patients with inter-renal (above 1 renal artery) clamping positions?
Which types of bridging stent grafts / Chimney stent grafts were used? Please add.
Table 4: Please add MACE to legend
Was there information in the included studies on whether these were all primary repairs or did they also include patients with revisions of prior failed endo repairs or proximal aneurysm progression after open repair? Please comment.
Discussion
As you state correctly, a relevant problem when comparing treatments of JAAA/PAAA is the heterogeneity of anatomical definitions in the literature. How were JAAA/PAAA in the included studies exactly defined? Maybe you can present this information in one of the tables in the results section.
You should point out that CHEVAR is usually only used for emergent/urgent procedures and not for elective patients, were FEVAR is preferred (you can cite the ESVS guidelines for this, recommendation 97). Please also dicuss the problem of gutter endoleaks in CHEVAR especially with >2 chmineys. Further you should discuss the issue that in FEVAR nowadays 4 fenestrations are usually preferred even for JAAA, so that the repair is often more proximal/extensive than it would have been the case in OSR.
5. Limitations:
As stated above, since 2010 endovacular experience, techniques and devices have improved a lot, so that you will still have an important learning curve bias and different generations of devices in your analysis.
Conclusion
One of your main results is a higher medium-term all-cause mortality after FEVAR compared to OSR. Despite the possible selection bias, which you describe correctly in the discussion, this should also be mentioned in the conclusion.
References:
A recently published review and network meta-analysis on the treatment of juxtarenal/short neck AAA by Patel et al. (Patel SR, Ormesher DC, Griffin R, Jackson RJ, Lip GYH, Vallabhaneni SR; UK-COMPASS Trial. Editor's Choice - Comparison of Open, Standard, and Complex Endovascular Aortic Repair Treatments for Juxtarenal/Short Neck Aneurysms: A Systematic Review and Network Meta-Analysis. Eur J Vasc Endovasc Surg. 2022 May;63(5):696-706. doi: 10.1016/j.ejvs.2021.12.042. Epub 2022 Feb 25. PMID: 35221243.) found no difference in midterm all cause mortality between OS, CHEVAR and FEVAR but also higher midterm reintervention rates after FEVAR. Although the inclusion criteria and treatment methods compared by Patel et al. were somewhat different from those in your work, you should incorporate this recent publication in your discussion.
Supplementary material:
In your search criteria you have also included „BEVAR“, but in the manuscript branched evar is not mentioned. Did any of the included studies also report on BEVAR as endovascular treatment or only FEVAR or maybe also customized grafts with both fenestrations and branches? Please clarify..
Author Response
We would like to thank the reviewers for the thorough and dedicated work they have put into reviewing our manuscript. We believe that their observations have improved the quality of our manuscript, and for this we would also like to thank them.
Reviewer 2:
Congratulations to the authors on this review comparing the medium-term outcomes of 3 treatment options for juxta- and pararenal aneurysms. The work is interesting and methodically sound; however, I have several questions and remarks.
ANS: We thank the reviewer for kind words.
Paragraph 1/Introduction:
The second sentence seems a bit unclear. What do you mean by „this category“ and „such patients“? Juxtarenal AAA? Hostile necks? Please clarify and rephrase.
ANS: Please excuse us for the misunderstanding. We were alluding to infrarenal AAA. This has been corrected.
Second paragraph: „JAAA defined by the ESVS….“ „are“ is missing.
ANS: We thank the reviewer for this observation. We have added the “ as needed.
Methods
Paragraph 2.1: What do you exactly mean by „Studies with standard, ruptured AAA…. were excluded from the analysis“? Do you mean infrarenal AAA? Were all infrarenal AAA excluded or only ruptured infrarenal AAA? What about emergent cases? And what AAA with a long enough but otherwise hostile neck, i.e. thrombus, calfication, conical shape, excessive angulation – I guess these were excluded as out of IFU? I guess that for OSR infected aneurysms were also excluded? Please specify your inclusion/exclusion criteria.
ANS: Please excuse us for the misunderstanding. We were alluding to infrarenal AAA. This has been corrected. All ruptured AAAs were excluded from the analysis as well because this is a special emergent cohort of patients that deserves another article. Also, we were strict with inclusion criteria, considering only studies reporting results of JAAA treatment, according to the definition of JAAA. Hence, studies reporting cases of patients with long but hostile necks have been excluded. Also, mycotic AAAs were excluded as well. Thank you for all of these suggestions, we explained this in more detail in the methodology.
Even by excluding studies published before 2010 you will still have different stentgraft generations!
ANS: We thank the reviewer for this comment. This is true, and while it would be impossible to account for these differences completely, in our approach we have sought to at least somewhat minimize this influence by including only newer studies. Clearly, in the period analyzed there could be overlapping studies where fenestrated stent grafts of different generations were used. However, we consider the last 12 years a reasonable period of time in which not only the latest generation devices were used, but there was undoubtedly a progressive improvement of the facilities, materials, and increasing experience.
How may visceral/renal bypasses in the OSR patients were there actually? Do the included studies report on this in detail? In my experience it is much more common to incorporate the renal arteries directly into the proximal anastomosis in open JAAA repair than by selective bypass. Therefore, the comparison to target vessel complications in FEVAR/CHEVAR is difficult. This aspect should also be added to your discussion.
ANS: Thank you for this question. Eight out of eleven studies reported the number of renal bypasses, not a single study reported that there was a need for SMA/CT bypass. Only one study reported results on pararenal AAA, the rest were juxtarenal AAAs. In total 269 renal bypasses/reattachments in 2334 patients (11.5%) were performed. We acknowledge this difficulty in the discussion.
Paragraph 2.3:
How was MACE defined?
ANS: MACE (major adverse cardiovascular event) was defined as a composite of cardiac death, myocardial infarction, coronary artery revascularisation, stroke, and hospitalization because of heart failure
The main issue with CHEVAR are gutter endoleaks. How were they defined? Did you count them as Type Ia? Please specify since this has impact on your results.
ANS: Thank you very much for your comment. Gutter endoleak was defined as persistent perigraft blood flow between parallel stent-grafts and the main body at the level of the proximal landing zone. As you know it represents a relatively frequent early occurrence after CHEVAR, but appears to resolve spontaneously in the majority of cases during early follow-up. Hence, we only include persistent type Ia endoleak.
Figure 1: You mention full text studies excluded due to lack of long term data – should this not be lack of midterm data since you report on midterm results?
ANS: We thank the reviewer for this observation. We have corrected the figure to state “not available medium-term data”:
Results
Paragraph 3.1:
Please add complete manufacturer data of all manufacturers (company, city, state/country).
ANS: All of the manufacturer data have been inserted accordingly.
Is there information in the included studies on wether any kind of renal protection (i.e. cold perfusion) was used in the patients with suprarenal clamping? Were there also patients with inter-renal (above 1 renal artery) clamping positions?
ANS: Thank you for your comment. Only three papers reported some kind of renal protection strategy; two used i.v. Manitol solution prior to and during clamping and one cold crystalloid solution (300ml per kidney). Yes, there were such patients with interrenal clamp position. We changed the results displayed in Table 2 accordingly.
Which types of bridging stent grafts / Chimney stent grafts were used? Please add.
ANS: Thank you for your question. The most commonly used bridging stent-graft in FEVAR and chimney graft in the ChEVAR group was Advanta V12 (Atrium Medical, Hudson, New Hampshire).
Table 4: Please add MACE to legend
ANS: We thank the reviewer for this observation. We have added the explanation for the abbreviation MACE to the legend.
Was there information in the included studies on whether these were all primary repairs or did they also include patients with revisions of prior failed endo repairs or proximal aneurysm progression after open repair? Please comment.
ANS: We thank the reviewer for this important remark. We included these patients as well. However, only five studies have these data, as you will see in the new column of Table 1. Unfortunately, due to a lack of reporting, we cannot distinguish whether these were revisions due to failed EVAR or proximal disease progression/pseudoaneurysm after OSR. We also cannot say whether these patients had worse outcomes since the results were not separately reported.
Discussion
As you state correctly, a relevant problem when comparing treatments of JAAA/PAAA is the heterogeneity of anatomical definitions in the literature. How were JAAA/PAAA in the included studies exactly defined? Maybe you can present this information in one of the tables in the results section.
ANS: Thank the reviewer for this suggestion. We have added the definitions of JAAA/PAAA as used in the studies in appendix G, seeing how there was no space for additional text in the tables in the results section.
You should point out that CHEVAR is usually only used for emergent/urgent procedures and not for elective patients, were FEVAR is preferred (you can cite the ESVS guidelines for this, recommendation 97). Please also dicuss the problem of gutter endoleaks in CHEVAR especially with >2 chmineys. Further you should discuss the issue that in FEVAR nowadays 4 fenestrations are usually preferred even for JAAA, so that the repair is often more proximal/extensive than it would have been the case in OSR.
ANS: Thank you for your excellent suggestions. We added a paragraph dedicated to the recent ESVS guidelines recommendations of the use of ChEVAR and FEVAR, as well as the importance of gutter endoleaks and the influence of more complex FEVAR devices in the occurrence of later complications.
- Limitations:
As stated above, since 2010 endovacular experience, techniques and devices have improved a lot, so that you will still have an important learning curve bias and different generations of devices in your analysis.
ANS: We thank the reviewer for this comment. We have added the following: “...but since the endovascular techniques have been further improving since then, it is possible that this bias is still present in our analysis.”
Conclusion
One of your main results is a higher medium-term all-cause mortality after FEVAR compared to OSR. Despite the possible selection bias, which you describe correctly in the discussion, this should also be mentioned in the conclusion.
ANS: We thank the reviewer for this important comment. We have updated the conclusion as following: “...and the possibility of existence of confounding by indication, bearing in mind the observational design of the included studies.”
References:
A recently published review and network meta-analysis on the treatment of juxtarenal/short neck AAA by Patel et al. (Patel SR, Ormesher DC, Griffin R, Jackson RJ, Lip GYH, Vallabhaneni SR; UK-COMPASS Trial. Editor's Choice - Comparison of Open, Standard, and Complex Endovascular Aortic Repair Treatments for Juxtarenal/Short Neck Aneurysms: A Systematic Review and Network Meta-Analysis. Eur J Vasc Endovasc Surg. 2022 May;63(5):696-706. doi: 10.1016/j.ejvs.2021.12.042. Epub 2022 Feb 25. PMID: 35221243.) found no difference in midterm all cause mortality between OS, CHEVAR and FEVAR but also higher midterm reintervention rates after FEVAR. Although the inclusion criteria and treatment methods compared by Patel et al. were somewhat different from those in your work, you should incorporate this recent publication in your discussion.
ANS: We thank the reviewer for mentioning this important publication. We have added a couple of sentences in the discussion mentioning this important NMA.
Supplementary material:
In your search criteria you have also included „BEVAR“, but in the manuscript branched evar is not mentioned. Did any of the included studies also report on BEVAR as endovascular treatment or only FEVAR or maybe also customized grafts with both fenestrations and branches? Please clarify..
ANS: Thank you for this comment. We wanted to scrutinize the literature for BEVAR, but there were no reports focusing specifically on JAAA/PAAA. BEVAR and combined FEVAR/BEVAR was used for thoracoabdominal and suprarenal AAA, which was out of the scope of this review.
Round 2
Reviewer 1 Report
The authors have sufficiently improved the manuscript.
Author Response
Reviewer #1:
The authors have sufficiently improved the manuscript.
ANS: We thank the reviewer for the comments, which have led to the improvement of the quality of our manuscript.
Reviewer 2 Report
Paragraph 2.1. Screening and study selection:
- Please insert „studies“ and „and“ as marked in the following in order to make this sentence clearer: „…., connective-tissue related aneurysms, studies containing less than 10 patients per treatment arm (to ensure enough experience in the treatment of this complex AAA pathology), performed before 2010 (to avoid the comparisons of different stent-graft generations), those analysing patients treated with infrarenal EVAR outside the instructions for use and those having less than one 6 month follow-up were excluded from the analysis.
- I agree with you about your answer to the following comment:
Even by excluding studies published before 2010 you will still have different stentgraft generations!
ANS: We thank the reviewer for this comment. This is true, and while it would be impossible to account for these differences completely, in our approach we have sought to at least somewhat minimize this influence by including only newer studies. Clearly, in the period analyzed there could be overlapping studies where fenestrated stent grafts of different generations were used. However, we consider the last 12 years a reasonable period of time in which not only the latest generation devices were used, but there was undoubtedly a progressive improvement of the facilities, materials, and increasing experience.
Nonetheless, I think you should rephrase the part „to avoid comparison of different stent-graft generations“. I suggest writing something like „to reduce the number of different stent-graft generations in the comparison“.
- Figure 1:
Thank you for adding information on the bridging stent grafts. There are some mistyping errors in this column (e.g. Benno et al.: Fluency, Chinsakchai et al: Begraft) Please review spelling in this column and correct accordingliy. Further, the explanation of BECSs (I guess balloon-expandable covered stents) is missing in the below-table legend; please add.
-
Author Response
Reviewer #2:
Paragraph 2.1. Screening and study selection:
- Please insert „studies“ and „and“ as marked in the following in order to make this sentence clearer: „…., connective-tissue related aneurysms, studies containing less than 10 patients per treatment arm (to ensure enough experience in the treatment of this complex AAA pathology), performed before 2010 (to avoid the comparisons of different stent-graft generations), those analysing patients treated with infrarenal EVAR outside the instructions for use and those having less than one 6 month follow-up were excluded from the analysis.
ANS: Thank you very much for your comment, we have corrected this in the manuscript.
- I agree with you about your answer to the following comment:
Even by excluding studies published before 2010 you will still have different stentgraft generations!
ANS: We thank the reviewer for this comment. This is true, and while it would be impossible to account for these differences completely, in our approach we have sought to at least somewhat minimize this influence by including only newer studies. Clearly, in the period analyzed there could be overlapping studies where fenestrated stent grafts of different generations were used. However, we consider the last 12 years a reasonable period of time in which not only the latest generation devices were used, but there was undoubtedly a progressive improvement of the facilities, materials, and increasing experience.
Nonetheless, I think you should rephrase the part „to avoid comparison of different stent-graft generations“. I suggest writing something like „to reduce the number of different stent-graft generations in the comparison“.
ANS: Thank you very much for your comment, we have corrected this in the manuscript.
- Figure 1:
Thank you for adding information on the bridging stent grafts. There are some mistyping errors in this column (e.g. Benno et al.: Fluency, Chinsakchai et al: Begraft) Please review spelling in this column and correct accordingliy. Further, the explanation of BECSs (I guess balloon-expandable covered stents) is missing in the below-table legend; please add.
ANS: Please excuse us for the spelling errors and thank you for noticing this. All English spelling mistakes have been corrected and an explanation for the term BECSs has been added to the table legend.